# Chiral Excitation of Exchange Spin Waves Using Gold Nanowire Grating

**Loic Temdie** [1,2], **Vincent Castel** [1,2], **Timmy Reimann** [3], **Morris Lindner** [3], **Carsten Dubs** [3], **Gyandeep Pradhan** [4], **Jose Solano** [4], **Romain Bernard** [4], **Hicham Majjad** [4], **Yves Henry** [4], **Matthieu Bailleul** [4] and **Vincent Vlaminck** [1,2,*]

1   Département Micro-Ondes, Institut Mines-Télécom (IMT) Atlantique, Technopole Brest-Iroise CS83818, CEDEX 03, 29238 Brest, France; loic.temdie-kom@imt-atlantique.fr (L.T.)
2   Laboratoire des Sciences et Techniques de l'Information de la Communication et de la Connaissance (Lab-STICC), Unité Mixte de Recherche (UMR) 6285, CNRS, Technopole Brest-Iroise CS83818, CEDEX 03, 29238 Brest, France
3   INNOVENT e.V. Technologieentwicklung, Pruessingstrasse 27B, 07745 Jena, Germany
4   Unité Mixte de Recherche (UMR) 7504, CNRS, Institut de Physique et Chimie des Matériaux de Strasbourg, Université de Strasbourg (IPCMS), CEDEX, 67000 Strasbourg, France
*   Correspondence: vincent.vlaminck@imt-atlantique.fr

**Abstract:** We propose an experimental method for the unidirectional excitation of spin waves. By structuring Au nanowire arrays within a coplanar waveguide onto a thin yttrium iron garnet (YIG) film, we observe a chiral coupling between the excitation field geometry of the nanowire grating and several well-resolved propagating magnon modes. We report a propagating spin wave spectroscopy study with unprecedented spectral definition, wavelengths down to 130 nm and attenuation lengths well above 100 μm over the 20 GHz frequency band. The proposed experiment paves the way for future non-reciprocal magnonic devices.

**Keywords:** spin waves; chirality; non-reciprocity; spectroscopy; thin YIG film

## 1. Introduction

Spin waves or their quanta magnons carry attractive properties for the development of wave-based computing technologies [1–5], in which the frequency and the phase of the waves can be used as new degrees of freedom for data processing. In particular, their ability to be stirred and shaped in the submicron scale [6–9] offer great perspectives for the miniaturization of interferometric devices. Furthermore, their integration is compatible with current CMOS [10] and surface acoustic wave nanotechnologies [11]. More significantly, chirality is a strong inherent property of magnetization dynamics [12–16], which puts the field of magnonics in the forefront for the integration of non-reciprocal microwave components such as isolators or circulators.

Recently, unidirectional transmission of exchange spin waves has been demonstrated by taking advantage of the chiral coupling between the dynamic dipolar field of resonating ferromagnetic nanowires and high wave vector spin waves in a thin YIG film [17–20]. This technique requires matching the ferromagnetic resonance (FMR) frequency of the ferromagnetic nanowires with the ones of the scanty propagating spin wave modes in the YIG film selected from the excitation geometry. This leads to a non-monotoneous excitation efficiency as each material inevitably has different magnetic properties, and thus different dispersion relations.

In this article, we present an alternative technique for the chiral excitation of exchange spin waves in a 55 nm thin YIG film using solely the microwave field of an Au nanowire array connected to a coplanar waveguide (CPW). The manuscript is organized as follows: Section 2 presents the design of the Au nanowire grating and the experimental protocol. In

Section 3, we show the unidirectional transmission spectra. In Section 4, we present the spectral analysis for all modes with the extracted group velocities and attenuation lengths in the 20 GHz frequency band and for the bias field ranging up to 500 mT.

## 2. Experimental Setup

### 2.1. Design of the Au Nanowire for the Chiral Excitation

The chiral excitation of spin waves from the microwave field of a nanowire essentially comes from the right-handedness of the magnetization precession at resonance. As sketched in Figure 1a, the phase profile of a propagating spin wave only matches the spatial distribution of microwave field $h$ of a nanowire for one direction of propagation, e.g., when cross-product $\vec{k} \times \vec{M}$ points along $+\vec{u}_x$ [21] (sketched in yellow in Figure 1a). Furthermore, the degree of chirality is all the more complete as the field lines are circular, and it gradually degrades as the aspect ratio thickness/width of the nanowire ($t_{Au}/w$) reduces [22]. This requires fabricating nanowires as thick as wide, which poses some challenge for nanofabrication.

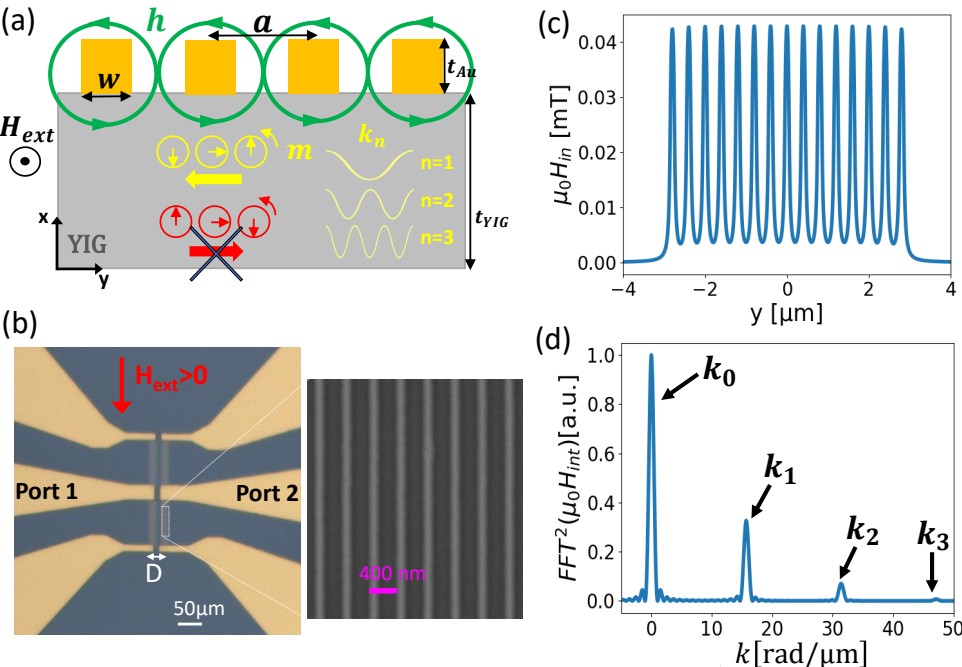

**Figure 1.** (**a**) Sketch of chiral excitation of SW from an Au nanowire grating. Green circle represents the microwave field $h$ of Au NW, yellow and red circle represents the magnetization dynamics $m$ respectively for left and right propagating direction. (**b**) Optical microscope image of the Au nanowire-grating sample. (**c**) Distribution of the in-plane microwave field. (**d**) Fourier transform of the in-plane exciting field.

We designed an array of 40 nm thick, 100 nm wide, and 35 μm- ong Au nanowires with a $a$ = 400 nm lattice constant, extending over 6 μm (e.g., $\simeq$15 wires). These gratings serve as spin wave antennas, and their spatial extension delimit the source and sink area. The extremities of the nanowires short-circuit the signal line and the ground lines of a CPW. In this way, we ensure that the fields of each wire are in phase, which improves the definition of the spin wave modes. Figure 1b shows an optical image of the Au nanowires grating sample obtained with a two-step fabrication process. In the first step, two nanowire arrays are at a distance of D = 20 μm, fabricated directly onto a 55 nm thin liquid phase epitaxial (LPE) YIG film [23] via electron beam lithography using an extra layer of conductive resist to circumvent the insulating nature of the substrate [24]. This step is followed by an e-beam evaporation of 40 nm of Au and a lift-off process. In the second step, a 80 nm thick Au CPW is aligned on top of the Au nanowires using optical lithography.

In order to assess the wave vectors excited from this Au-grating, we consider the in-plane component of the microwave field of the nanowire array as shown in Figure 1c. We used the expression of the Oersted field produced by a single wire with a rectangular section [25], and considered a uniform current of 0.01 mA in each wire, consistent with the usual values of microwave input power of P = −20 dBm and a CPW impedance of R $\simeq$ 30 Ω. We then plot the square of the Fourier transform of this field distribution as shown in Figure 1d, which represents the coupling efficiency in a reciprocal space. With such a geometry, we obtain a clear comb of wavevectors that are multiples of the inverse of the grating periodicity, namely $k_n = n2\pi/a$. Although the amplitudes of the higher-frequency peaks appear to decay rather rapidly, this spin wave antenna geometry provides us with four well-defined, evenly spaced, and workable modes in a broad range of a wavevector of [0–50] rad·μm$^{-1}$.

### 2.2. Measurement Protocol

We performed spin wave spectroscopy measurement on the fabricated sample which was placed into a homemade electromagnet mounted onto a Karl Süss PM-8 microwave probe station. We connected the sample with GGB picoprobes to a vector network analyzer Rohde & Schwarz ZNA-43GHz. Due to the relatively small amplitude of the spin wave signal, we always proceed to a relative measurement, in which we subtract a reference measurement at a different static magnetic field $H_{ref}$ far enough that it does not interfere in the frequency span with the resonant signal measured at $H_{res}$. Finally, as the nanowire array coupled inductively to the spin wave, we represented our relative measurements in units of inductance [26,27]:

$$\Delta L_{ab}(f, H) = \frac{1}{i\,2\pi\,f}(Z_{ab}(f, H) - Z_{ab}(f, H_{ref})),\tag{1}$$

where subscripts $(a, b)$ denote either a transmission measurement from ports $b$ to port $a$, or a reflection measurement performed on the same port if $a = b$.

### 3. Unidirectional Excitation of Spin Waves

Figure 2a shows a $(H_{ext}, f)$ mapping of the reflection spectra $\Delta L_{11}$ acquired between −192 mT and 192 mT. We detect four well-separated branches corresponding to some of the various excited spin wave wavevectors to be identified, the lowest of which corresponds to the $k_0 = 0$ FMR mode. A fit of the $k_0$ branch using the Kittel relation [28] gives us a gyromagnetic ratio $\gamma/2\pi = 28.2 \pm 0.2$ GHz·T$^{-1}$ and an effective magnetization $\mu_0 M_{eff} = 183 \pm 3$ mT, in good agreement with previous characterization on similar films [19], and suggesting no in-plane anisotropy. We show in Figure 2b,c the $(H_{ext}, f)$ mapping of the transmission spectra $\Delta L_{12}$ and $\Delta L_{21}$ measured at the −10 dBm input power for which we purposely saturated the color scale in order to reveal most of the detectable propagating modes. We also show in Figure 2d–i the zoom of the spectra at 65 mT for the six detected peaks measured at different power to boost the signal-to-noise ratio while ensuring a linear response regime. The spectra of Figure 2d,e were measured at −20 dBm, the spectrum of Figure 2f at 0 dBm, and the ones of Figure 2g–i at +10 dBm. When adjusting the power for each peak, we carefully checked that the spectra remain symmetrical and do not shift in frequency, confirming that we remain far below the non-linear threshold. Each spectra reveal clear features of typical propagating spin wave measurement, namely oscillatory signal whose envelope reproduces the spectral signature of Figure 1d. Furthermore, in comparison with regular CPW antennas [7,20,27,29], the Au-grating techniques offer an unprecedented spectral definition, with well-resolved, narrower and symmetrical spectra of six distinct spin wave modes ranging over a 10 GHz bandwidth at any given field.

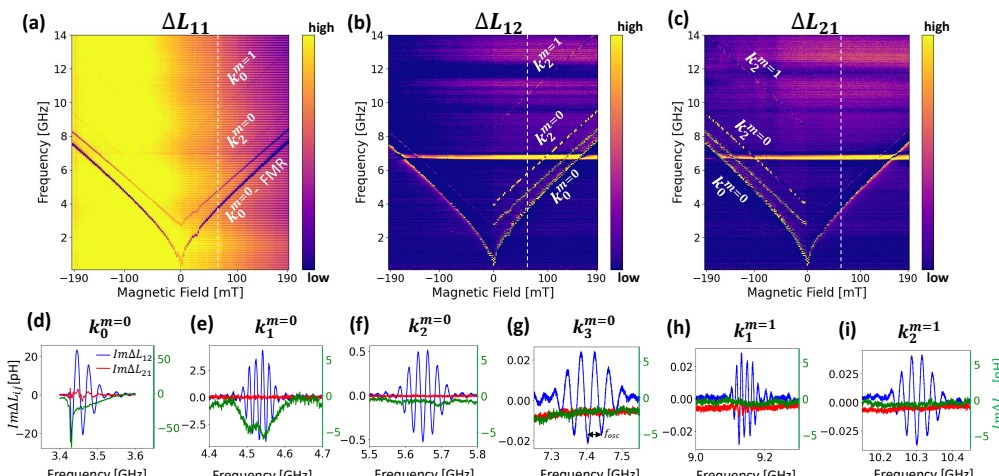

**Figure 2.** Mapping ($H_{ext}$, $f$) of (**a**) the reflection spectra, and (**b**,**c**) the transmission spectra measured from −192 mT to 192 mT. Spectra obtained at +65 mT for (**d**–**g**) the first four spin wave branches corresponding to the uniform thickness mode $m = 0$, and (**h**,**i**) the next two branches to the first perpendicular standing spin wave mode (PSSW) $m = 1$ Blue (resp. red) lines are $\Delta L_{12}$ (resp. $\Delta L_{21}$), and the green lines the $\Delta L_{11}$.

On the one hand, we observe a partial non-reciprocal transmission for the first and the fifth observed branches (Figure 2h). This partial chirality would most likely be improved with thicker nanowires, the current aspect ratio $t/w$ being less than 0.5 in our sample. On the other hand, the transmission for all the other branches is perfectly unidirectional. Furthermore, the non-reciprocity is reversed upon switching the polarity of the applied static field, namely for positive field values. The transmission occurs from Port 2 to Port 1, and, conversely, the spin waves only propagate from Port 1 to Port 2 for negative fields.

In addition, we also observe that the relative amplitudes of the first four resonance peaks agree relatively well with the spectral definition given by the Fourier transform of this microwave field distribution. Lastly, the period of oscillation $f_{osc}$ of the spectra increases gradually between the second and the fourth peak, which appears to be the largest of all. The fifth peak appears to have the smallest period of oscillation, which suggests it is a higher-order perpendicular standing spin wave mode (PSSW).

## 4. Modal Analysis of the Propagating Spin Waves

### 4.1. Wave Vector Identification

The remarkable spectral definition offered by this technique makes it a technique of choice for broadband spin wave spectroscopy characterization. In order to identify and analyze the propagation properties of all the detected peaks, we track the field dependence of the resonant frequencies, periods of oscillation, and amplitudes for each branch. Firstly, we proceed to a graphical verification of the wave vector for each field value by matching the Kalinikos–Slavin dispersion relations $f_{res}(k, H_{ext})$ [30] with the peak position using the magnetic parameters obtained from the FMR and an exchange constant $A_{ech} = 3.85$ pJ.m$^{-1}$ based on a prior study [19]:

$$\omega_m^2(k_m) = (\omega_H + \eta k_m^2)(\omega_H + \eta k_m^2 + F_m \omega_M),$$ (2)

where $\omega_H = \gamma \mu_0 H_{ext}$, $\omega_M = \gamma \mu_0 M_s$, $\eta = \frac{2\omega_M A_{ech}}{\mu_0 M_s^2}$, and $k_m^2 = k^2 + (\frac{m\pi}{t})^2$ the wave vector modulus squared of the m$^{th}$-PSSW mode. We represent in Figure 3a the dispersion of both the uniform thickness mode ($m = 0$) and the first PSSW ($m = 1$) mode at 270 mT, as well as their intersections with the peak positions to identify the corresponding wavevector. Following this methodology and taking an average for each mode, we assign the first four propagating peaks to the uniform thickness modes ($m = 0$), and the last two to the

first PSSW mode ($m = 1$). The obtained values of wavevector, which are summarized in Table 1, come very close to the expected $k_n = n\frac{2\pi}{a}$ spectral definition. Figure 3b also shows an excellent agreement between the measured field dependence of the resonant frequency with the dispersion relation Equation (2) for all modes. Furthermore, we notice that although the first PSSW $k_0^{m=1}$ mode ($n = 0$, $m = 1$) clearly appears in the reflection spectra (Figure 2a), it is not visible in the transmission spectra. Indeed, the slope of the dispersion relation close to $k = 0$ for $m = 1$ is zero, implying a null group velocity at a low wavevector.

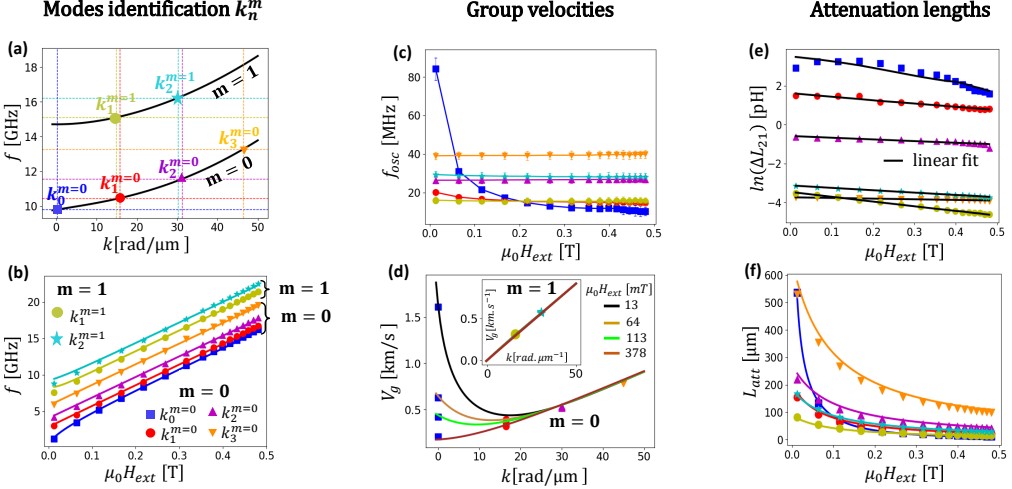

**Figure 3.** (**a**) Dispersion relation for the uniform ($m = 0$) and the first PSSW ($m = 1$) modes at $\mu_0 H_{ext}$ = 270 mT applied field. (**b**) Field dependence of the resonance frequency for all resonance peaks. (**c**) Field dependence of the period of oscillation $f_{osc}$. (**d**) Comparison of the measured group velocities with theoretical expression at several fields. (**e**) Spectra amplitude dependence with the normalized frequency. (**f**) Field dependence of the attenuation length for all wave vectors. Solid lines are the calculated attenuation length $L_{att} = \frac{2v_g}{\alpha(2\omega_H + \omega_M)}$.

**Table 1.** Summary of the modal analysis of the transmission spectra, their respective group velocity, and Gilbert damping.

| Modes $(n, m)$ | $n\frac{2\pi}{a}$ | $k$ [rad·μm$^{-1}$] | $v_g^{max}$ [m·s$^{-1}$] | $\alpha$ $10^{-4}$ |
|---|---|---|---|---|
| (0,0) | 0 | $0.03 \pm 0.02$ | 1614 * | $1.7 \pm 0.1$ |
| (1,0) | 15.7 | $16.5 \pm 0.5$ | 397 | $1.4 \pm 0.2$ |
| (2,0) | 31.4 | $31.8 \pm 0.4$ | 535 | $1.3 \pm 0.1$ |
| (3,0) | 47.1 | $47.0 \pm 0.2$ | 791 | $0.8 \pm 0.1$ |
| (1,1) | 15.7 | $15.8 \pm 0.8$ | 322 | $2.1 \pm 0.2$ |
| (2,1) | 31.4 | $30.4 \pm 0.7$ | 587 | $1.9 \pm 0.1$ |

* value obtained at $\mu_0 H_{ext}$ = 13 mT.

### 4.2. Group Velocity and Attenuation Length

The oscillatory signature of the transmission spectra occurs due to the non-purely monochromatic nature of the excitation (e.g., the finite width in the peaks of the Fourier transform Figure 1d), and the gradual dephasing accumulated over the propagation between the excited wavelengths within a single peak. From the period of oscillation $f_{osc}$, we can estimate the group velocity of the mode according to $v_g = f_{osc} * D$ [7,27], where $D$ = 20 μm is the propagation distance. We present in Figure 3c the field dependence of this period of oscillation for all wavevectors $(n, m)$. We observe a fast decay of $f_{osc}$ with the field only for $k_0^{m=0}$, while it remains fairly constant for the other wavevector. We summarize in Figure 3d the wavevector dependence of the obtained group velocities at representative

field values across the full measurement range. We obtain an excellent agreement with the calculated group velocities from the derivative of the dispersion relations $v_g(H_{ext}, k) = \frac{d\omega_m}{dk}$. We summarize the measured group velocities for all modes in Table 1. In addition, one can see from the shape of $v_g(k)$ that the higher wave vectors ($k_2, k_3$) are already in the exchange regime of the spectrum.

Lastly, we assess the field dependence of the attenuation lengths for all peaks. We consider an exponential decay of the spin wave (namely $\Delta L_{12} \approx Ae^{-\frac{D}{L_{att}}}$) for which we opt for the low-wavevector approximation of the attenuation length $L_{att} = \frac{2v_g}{\alpha(2\omega_H + \omega_M)}$, where $\alpha$ is the Gilbert damping. We plot in Figure 3e the evolution of the logarithm of each peak amplitudes with the field. We obtain clear linear dependence, from which we extract an effective Gilbert damping constant for each wave vector that we summarize in Table 1. While we obtain similar values with the tabulated Gilbert damping in high quality thin LPE YIG obtained from FMR measurement [23], we seem to observe a slight reduction in the effective damping constant with the wavevector for the uniform thickness mode $m = 0$. In addition, the two branches of the PSSW ($m = 1$) mode display a slightly higher damping constant. Finally, in Figure 3d we plot the field dependence of the obtained attenuation length for all modes and verify a good agreement with the low-wavevector approximation across the full field range. We note that the higher wave vector $k_3^{m=0}$ has an attenuation length well above 100 μm over the whole 20 GHz frequency band, suggesting possible propagation over macroscopic distances.

## 5. Conclusions

In conclusion, we showed that a grating of nanowires can excite unidirectional exchange spin waves in an extended in-plane magnetized thin YIG film. We observed a partial chirality at a wave vector lower than 16 rad·μm$^{-1}$, while it is perfect for higher wavevectors. We expected that the partial chirality at a lower wave vector would be improved with thicker nanowires. The proposed technique provides an unprecedented spectral definition with evenly spaced and well-resolved spin wave modes of up to 50 rad·μm$^{-1}$ ranging over a 10 GHz bandwidth at any applied field of up to 500 mT. The comb of the wave vector could even be broadened up to 100 rad·μm$^{-1}$ by reducing the nanowire periodicity by a factor of two. Therefore, this constitutes a method of choice for broad band high wave vector spin wave spectroscopy studies. In essence, this technique can be applied to any orientation of the external magnetic field, and the spectral distribution could be engineered with the grating dimension to adjust the multi-mode bandwidth operation. Our findings have important implications for the development of non-reciprocal magnonic devices.

**Author Contributions:** Conceptualization, V.V., L.T. and M.B.; sample fabrication, L.T., J.S., R.B. and H.M.; methodology, V.V. and L.T.; validation, V.V., L.T., M.B. and Y.H.; formal analysis, L.T. and V.V.; investigation, V.V., L.T. and G.P.; resources, V.V.; data curation, L.T.; writing—original L.T. and V.V.; writing—review and editing, L.T., V.V., V.C., Y.H. and M.B.; visualization, L.T. and V.V.; supervision, V.V.; project administration, V.V.; funding acquisition, V.V.; T.R., M.L. and C.D. provided the YIG film. All authors have read and agreed to the published version of the manuscript.

**Funding:** This work was supported by the French National Research Agency (ANR) under the project MagFunc, the Region Bretagne with the CPER-Hypermag project, and the Département du Finistère through the project SOSMAG. M.L. was funded by the German Bundesministerium für Wirtschaft und Energie (BMWi) under Grant No. 49MF180119.

**Institutional Review Board Statement:** Not applicable.

**Informed Consent Statement:** Not applicable.

**Data Availability Statement:** The data that support the findings of this study are available from the corresponding author upon reasonable request.

**Acknowledgments:** We want to acknowledge support from Bernard Abiven for the elaboration of the electromagnet. C.D. thanks O. Surzhenko for magnetic characterisation and R. Meyer for technical assistance.

**Conflicts of Interest:** The authors declare no conflict of interest.

## Abbreviations

The following abbreviations are used in this manuscript:

| | |
|---|---|
| YIG | Yttrium Iron Garnet |
| CMOS | Complementary metal–oxide semiconductor |
| FMR | Feromagnetic resonance |
| LPE | Liquid Phase Epitaxy |
| CPW | Coplanar Waveguide |
| PSSW | Perpendicular Standing Spin Wave |

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
