# Peer review of "Chiral Excitation of Exchange Spin Waves Using Gold Nanowire Grating"

_magnetochemistry, doi:10.3390/magnetochemistry9080199_

Round 1

Reviewer 1 Report

The research paper, "Chiral Excitation of Exchange Spin Waves Using Gold Nanowire Grating," introduces a pioneering experimental approach for instigating unidirectional excitation of spin waves. By incorporating arrays of gold (Au) nanowires within a coplanar waveguide on a thin yttrium iron garnet (YIG) film, the authors have enabled a chiral coupling between the excitation field geometry of the nanowire grating and several prominent propagating magnon modes.

Significantly, this work propels the understanding and application of spin wave excitation to new heights, potentially catalyzing advancements within the spheres of magnonics and non-reciprocal microwave components.

However, I believe the paper could benefit from a more comprehensive review of the existing literature. There are numerous studies describing analogous techniques, and a more detailed comparison of their concept with other coplanar waveguide (CPW) antennas would enhance the overall clarity and depth of the paper.

One suggestion I would like to offer is the potential inclusion of simulations using software tools such as Comsol, Lumerical, or CST to calculate the precise spatial distribution of the magnetic field produced by this antenna. Figure 1b and 1d raise some doubts regarding their construction, and full simulations would provide a more robust and accurate depiction of the field distribution, allowing for better optimization of the antenna design.

Additionally, attention should be directed towards rectifying the inconsistencies observed across the figures, including variable sizes and font styles. These discrepancies detract from the paper's overall aesthetic quality. Citation corrections are also warranted: citation 10 is missing due to a LaTeX error, and the subsequent Electra92 reference employs a word-based style that deviates from the standard numerical format.

Despite these critiques, the paper stands out due to its well-structured argumentation and comprehensive detailing of the study. Its relevance and potential contributions to the scientific community are evident, and I therefore recommend its publication in Magnetochemistry in its present state.

Author Response

The research paper, "Chiral Excitation of Exchange Spin Waves Using Gold Nanowire Grating," introduces a pioneering experimental approach for instigating unidirectional excitation of spin waves. By incorporating arrays of gold (Au) nanowires within a coplanar waveguide on a thin yttrium iron garnet (YIG) film, the authors have enabled a chiral coupling between the excitation field geometry of the nanowire grating and several prominent propagating magnon modes.

Significantly, this work propels the understanding and application of spin wave excitation to new heights, potentially catalyzing advancements within the spheres of magnonics and non-reciprocal microwave components.

However, I believe the paper could benefit from a more comprehensive review of the existing literature. There are numerous studies describing analogous techniques, and a more detailed comparison of their concept with other coplanar waveguide (CPW) antennas would enhance the overall clarity and depth of the paper.

  • We added a sentence comparing the definition, their symmetry and width of the spectra obtained with the Au-grating technique, compared with usual CPW antennas: "Furthermore, in comparison with regular straight CPW antennas \cite{Loayza2018,Vlaminck2010,Gladii2_2016, Li2023}, the Au-grating techniques offers an unprecedented spectral definition, with well-resolved, narrower and symmetrical spectra of 6 distinct spin wave modes ranging over a 10\,GHz bandwidth at any given field."

One suggestion I would like to offer is the potential inclusion of simulations using software tools such as Comsol, Lumerical, or CST to calculate the precise spatial distribution of the magnetic field produced by this antenna. Figure 1b and 1d raise some doubts regarding their construction, and full simulations would provide a more robust and accurate depiction of the field distribution, allowing for better optimization of the antenna design.

  • In a previous article (“Spin wave diffraction model for perpendicularly magnetized films”, DOI: 10.1063/5.0128666), we did compare CST simulations of microwave fields of spin wave antenna with our approximated expression of the Oersted field of an infinitely straight rectangular conductor varying dc current. Both field are in fact almost identical. Since the length of the nanowire (35µm) is much smaller than the electromagnetic wavelength of the microwave power flowing in the CPW, it is fair to say that the excitation field is in phase in the whole length of the nanowire. This justify our approximation.

Additionally, attention should be directed towards rectifying the inconsistencies observed across the figures, including variable sizes and font styles. These discrepancies detract from the paper's overall aesthetic quality. Citation corrections are also warranted: citation 10 is missing due to a LaTeX error, and the subsequent Electra92 reference employs a word-based style that deviates from the standard numerical format.

  • We improved theses points.

Despite these critiques, the paper stands out due to its well-structured argumentation and comprehensive detailing of the study. Its relevance and potential contributions to the scientific community are evident, and I therefore recommend its publication in Magnetochemistry in its present state.

Reviewer 2 Report

The authors report nonreciprocal spin wave propagation using chiral excitation from a set of nanostructured wire antennas. The experimental work is well designed and conducted, with clear evidence of nonreciprocal transmission behavior observed in the dataset. The overall presentation of the work is good, except a few comments that the authors should correct and improve, in a resubmitted version. 

1. Clearly, the authors did not proof-read the manuscript well, there are Latex errors with Refs here and there: Page 1, line 13, missing Ref. 8. And page 2, line 57.

2. Figure 1b, it seems an G-S-G configuration is used instead of a conventional G-S. The authors should clearly indicate which contact is what (signal line in the very middle?), and explain why such GSG is used in lieu of GS. And if using GS, would the conclusion be perhaps different? 

3. The spacing parameter, D, was introduced but not discussed. What would be the range of D for observing good nonreciprocal effect in such YIG devices?

4. What is the microwave loss at the antenna by using such nanowires, as compared to a regular IDT or waveguide? I hope the the loss is not too large? 

5. The dataset provides clear evidence for the effect and support the conclusion. However, the theoretical address is too brief and need improvement. It is perceivable that the phase matching is important, but detailed analysis and design rule suggestion were not given. Having a ratio of t/w is a bit too superficial. The authors should spend some more words and elucidate how the various dimensions of the antenna design change their coupling to what relevant modes, etc. This would greatly benefit the audience, which I bet is what the authors would be in favor of doing.  

6. Related to the above comments, the YIG used in this study is relatively thin. If using thicker YIG, more higher order modes would possibly be induced? Will the effect be enhanced or attenuated as the mode indices increase? Some theoretical modeling or even simulations might come handy for this particular point. 

6. Figure 3 currently is too dense and the figure legends are too small, hard to read. Please consider separating figures or simplifying. 

English is fine. 

Author Response

The authors report nonreciprocal spin wave propagation using chiral excitation from a set of nanostructured wire antennas. The experimental work is well designed and conducted, with clear evidence of nonreciprocal transmission behavior observed in the dataset. The overall presentation of the work is good, except a few comments that the authors should correct and improve, in a resubmitted version. 

  1. Clearly, the authors did not proof-read the manuscript well, there are Latex errors with Refs here and there: Page 1, line 13, missing Ref. 8. And page 2, line 57.

-> I do not have this error when compiling the Latex document using overleaf, nor on the preprint version (doi: 10.20944/preprints202306.1414.v1). The error must come with the compilation from MDPI.

  1. Figure 1b, it seems an G-S-G configuration is used instead of a conventional G-S. The authors should clearly indicate which contact is what (signal line in the very middle?), and explain why such GSG is used in lieu of GS. And if using GS, would the conclusion be perhaps different? 

-> We added the letters G-S-G to the contacts in the figure. GSG configuration is the one that respect the most the cylindrical symmetry from the coaxial cables, therefore it should always be preferred to a GS configuration. Nevertheless, using a GS geometry would work exactly the same way, as each Au gratings connects each Gs to the S, e.g. we have two gratings in our case while there would be only one with a GS configuration.

  1. The spacing parameter, D, was introduced but not discussed. What would be the range of D for observing good nonreciprocal effect in such YIG devices?

-> The spacing parameter D is not a relevant parameter for the non-reciprocity, which is solely due to the chiral coupling between the microwave field of the Au grating and the YIG film. We introduced the parameter D to evaluate the group velocity, and the attenuation length in Fig. 3. As long as the propagation distance D is smaller the attenuation length, very little decrease in amplitude are to be expected. For the distance D greater than Latt, the signal will start to drop significantly.     

  1. What is the microwave loss at the antenna by using such nanowires, as compared to a regular IDT or waveguide? I hope the the loss is not too large? 

-> Ohmic losses are essentially taken by the nanowire gratings, which are short-circuiting Ground and Signal lines. The real part of our antennas impedance is around 30 Ohms for the whole frequency range. Although, impedance matching for spin wave antenna could certainly be improved further, we know from experience that it is not so important as long as the impedance remains under fairly 150 Ohms, as we always resort to a relative measurement (DZij=Zij(Hres)- Zij(Href)). In the case of larger ohmic losses, for instance if the grating thickness is reduced, it will not change the conclusions of this study. 

  1. The dataset provides clear evidence for the effect and support the conclusion. However, the theoretical address is too brief and need improvement. It is perceivable that the phase matching is important, but detailed analysis and design rule suggestion were not given. Having a ratio of t/w is a bit too superficial. The authors should spend some more words and elucidate how the various dimensions of the antenna design change their coupling to what relevant modes, etc. This would greatly benefit the audience, which I bet is what the authors would be in favor of doing.  

-> The length of the nanowire (35µm) is much smaller than the electromagnetic wavelength of the microwave power flowing in the CPW. Therefore, it is fair to say that the excitation field is in phase in the whole length of the nanowire.

 Chirality in magnetization dynamics has been largely documented and explained. In the case of uniform dynamics, it directly results from the gyrotropic nature of the dynamic susceptibility which display two eigen modes with circular polarization, with only the right-handed polarization having a resonant behavior. We added the reference of the A.G. Gurevich textbook for the non-specialized public to acquire the necessary knowledge on magnetization dynamics. We complemented our formulation of the chiral mechanism with the following selection rule:

As sketched in Figure 1-(a), the phase profile of a propagating spin wave only matches the spatial distribution of the microwave field h of a nanowire for one direction of propagation, e.g. when the cross-product k*M points along +ux (sketched in yellow in Figure 1-(a)).

Changing the lattice spacing will space further apart the selected peak kn according to the selection rules kn=n*2p/a. This point was clearly stated in line 62, as well as in the conclusion line 167. The density of nanowire array will not affect the chirality of the excitation, and even in the case of a single nanowire, it was shown to excite unidirectional spin wave (cf ref. 18 : https://doi.org/10.1007/s12274-020-3251-5).

  1. Related to the above comments, the YIG used in this study is relatively thin. If using thicker YIG, more higher order modes would possibly be induced? Will the effect be enhanced or attenuated as the mode indices increase? Some theoretical modeling or even simulations might come handy for this particular point. 

-> Indeed, already at this somewhat thin thickness of 55nm, the first higher order thickness modes m=1, e.g. Perpendicular Standing Spin Wave (PSSW) mode are already being observed, and analyzed consistently with Kalinikos-Slavin theory of spin wave. We referred to them as k0m=1, k1m=1 and k2m=1. Namely, we showed in agreement with Kalinikos-Slavin theory that k0m=1 is absent of the transmission spectra as its group velocity is zero, and the next two modes k1m=1 and k2m=1 also display unidirectional transmission (see figure 2-(h) and 2-(i)). Thicker films would bring the higher order modes closer in frequency, which could complicate the analysis if they overlap.

  1. Figure 3 currently is too dense and the figure legends are too small, hard to read. Please consider separating figures or simplifying. 

-> We increased the size of the legends in figure 3, spaced them apart, and added three labels in order for the reader to read it as three distinct columns: the first one for the identification of the wavevector, the second one for the evaluation of the group velocities, and the third one about the attenuation of each modes.

Reviewer 3 Report

The manuscript demonstrates an experiment on precize adjusting paramters of propagating spin waves corresponding to perpendicular standing spin wave modes in narrow areas of the sample. The experimental sample is manufactured of an insulating ferromagnetic plate and metalic-nanowire grating with nanowires aligned to the propagation direction. The authors report on controlable unidirectional propagation of spin waves with extremely high spectral resolution under the application of a constant bias field and AC Oerstead field from the nanowires. In my opinion, the concept is of potential importance for the magnonics community. The manuscript is well written, however, several points require to be addressed..

1. I suggest, for clarifying the presentation, to write down explicitely that the gratings serve as antenas and they belong to spatial area of the spin wave source/sinc, (if I understand well). Looking at Fig. 1(b), one can think that a single grating elongates through whole the sample... Moreover, according to Fig. 1, the nanowires cannot be of only 35nm long, as claimed.

2. At the end of page 3, it is claimed that the condition of linear response of the system is insured. Such a claim requires a confirmation via estimating the driving Oerstead field and its comparison to other characteristic fields. Avoiding nonlinearity is important when evaluating the Gilbert damping constant (page 5).

3. Concluding (page 6) on the demonstration of "unprecedented spectral definition with evenly spaced and well-resolved spin wave modes", a  comparison to the state of the art is in order.

Author Response

The manuscript demonstrates an experiment on precize adjusting paramters of propagating spin waves corresponding to perpendicular standing spin wave modes in narrow areas of the sample. The experimental sample is manufactured of an insulating ferromagnetic plate and metalic-nanowire grating with nanowires aligned to the propagation direction. The authors report on controlable unidirectional propagation of spin waves with extremely high spectral resolution under the application of a constant bias field and AC Oerstead field from the nanowires. In my opinion, the concept is of potential importance for the magnonics community. The manuscript is well written, however, several points require to be addressed..

  1. I suggest, for clarifying the presentation, to write down explicitely that the gratings serve as antenas and they belong to spatial area of the spin wave source/sinc, (if I understand well). Looking at Fig. 1(b), one can think that a single grating elongates through whole the sample... Moreover, according to Fig. 1, the nanowires cannot be of only 35nm long, as claimed.

-> Ok, we explicitely wrote that the gratings serve as antennas. Thank you for the tipo, it is of course 35µm-long wire and not 35nm.

  1. At the end of page 3, it is claimed that the condition of linear response of the system is insured. Such a claim requires a confirmation via estimating the driving Oerstead field and its comparison to other characteristic fields. Avoiding nonlinearity is important when evaluating the Gilbert damping constant (page 5).

-> When reaching power close to the non-linear threshold, the first modification to check is the distortion and frequency shift of the spectra, which is due to a foldover type of behavior. We carefully adjusted the power for each mode to improve signal to noise ratio, while ensuring that the spectra of each mode remain symmetrical, and do not shift in frequency. Therefore, it is safe to say that the measurements were done in the linear regime. We added the following sentence: “When adjusting the power for each peaks, we carefully checked that the spectra remain symmetrical, and do not shift in frequency, confirming that we remain far below non-linear threshold.”

  1. Concluding (page 6) on the demonstration of "unprecedented spectral definition with evenly spaced and well-resolved spin wave modes", a comparison to the state of the art is in order.

-> We reformulated our statement to compare our results with regular straight CPW antennas, which have a much broader spectral width (Dk), and therefore result in a broader linewidth of the spectra in frequency. We added the following statement: “Furthermore, in comparison with regular straight CPW antennas \cite{Loayza2018,Vlaminck2010,Gladii2_2016, Li2023}, the Au-grating techniques offers an unprecedented spectral definition, with well-resolved, narrower and symmetrical spectra of 6 distinct spin wave modes ranging over a 10\,GHz bandwidth at any given field.”.

Reviewer 4 Report

The manuscript describes a method and its experimental
implementation enabling a unidirectional excitation of spin waves
using gold nanowire grating. I find the proposed method and
reported experimental results interesting and potentially useful
and, thus, worth of publication in Magnetochemistry.

Anyway, I think the presentation of the method can be improved,
especially for those Readers who are nonspecialists, for example:

* It would be meaningful to give a simple physical explanation why
gold nanowire grating is so improves for the proposed method
compared to other related methods.

* Explaining the physical meaning of various quantities, e.g.,:
\Delta L_{ab} and Z_{ab} in Eq. (1), and M_s, \omega_H,
\omega_M, and \eta in Eq. (2) could be useful.

* The meaning of \alpha (that denotes Gilbert damping) and v_g^{max}
(corresponding to the maximal group velocity) could be added the caption
of Table 1.

* The period of oscillations is denoted by f_{osc} and shown in Fig.
2(d). This notation is quite confusing for me, because symbol f
usually denotes frequency rather than a period (which is typically
denoted by T).

* It is mentioned in the abstract about "attenuation lengths ...
[ranging] over the 20GHz frequency band", and in the conclusions
it is written about "spin wave modes ranging ... over a 10GHz
bandwidth." For clarity I would suggest to write about these
ranges both in the Abstract and Conclusions. One should not be
afraid of such repetitions, if they improve the clarity of the
paper and explain the applicability of the method.

* Below Eq. (2), "vector modulus" should read "vector modulus
squared".

* There is a typo in line 13 related to (probably) a wrong label of
Ref. [8] in the source file.

* The list of abbreviations at the end of the article could be
enlarged by including, e.g., PSSW, which stands for a
perpendicular standing spin wave mode.

No comments

Author Response

The manuscript describes a method and its experimental
implementation enabling a unidirectional excitation of spin waves
using gold nanowire grating. I find the proposed method and
reported experimental results interesting and potentially useful
and, thus, worth of publication in Magnetochemistry.

Anyway, I think the presentation of the method can be improved,
especially for those Readers who are nonspecialists, for example:

* It would be meaningful to give a simple physical explanation why
gold nanowire grating is so improves for the proposed method
compared to other related methods.

-> The Au-grating method offers a much better spectral definition up to 50 rad/µm, which is about 5 times the usual limit with usual CPW antenna. This point was emphasized along the text, as well as in the conclusion.

* Explaining the physical meaning of various quantities, e.g.,:
\Delta L_{ab} and Z_{ab} in Eq. (1), and M_s, \omega_H,
\omega_M, and \eta in Eq. (2) could be useful.

-> We are sorry that the referee ask to rephrase the meaning of these four terms. This is the clearer formulation that has been widely adopted by several groups over the years. We regret that we could hardly make it clearer in the text. 

* The meaning of \alpha (that denotes Gilbert damping) and v_g^{max}
(corresponding to the maximal group velocity) could be added the caption
of Table 1.

-> ok thanks.

* The period of oscillations is denoted by f_{osc} and shown in Fig.
2(d). This notation is quite confusing for me, because symbol f
usually denotes frequency rather than a period (which is typically
denoted by T).

-> We agree that a period of oscillation in unit of frequency may sound misleading. However, we refer here to oscillatory nature of spin wave spectra acquired through a frequency sweep. We keep this notation as it is the one that is commonly used (cf other publication on the spin wave spectroscopy).

* It is mentioned in the abstract about "attenuation lengths ...
[ranging] over the 20GHz frequency band", and in the conclusions
it is written about "spin wave modes ranging ... over a 10GHz
bandwidth." For clarity I would suggest to write about these
ranges both in the Abstract and Conclusions. One should not be
afraid of such repetitions, if they improve the clarity of the
paper and explain the applicability of the method.

->The 10 GHz bandwidth refers to frequency difference between first and last detected modes at a given field. We added “at any given field” in the text. The 20 GHz frequency band refer to how this 10GHz bandwidth is moved with the applied bias field in our study.

* Below Eq. (2), "vector modulus" should read "vector modulus
squared".

-> Ok thanks.

* There is a typo in line 13 related to (probably) a wrong label of
Ref. [8] in the source file.

-> This is strange as the error does not show up when I compile the Latex document.

* The list of abbreviations at the end of the article could be
enlarged by including, e.g., PSSW, which stands for a
perpendicular standing spin wave mode.

-> ok Thanks.

Round 2

Reviewer 3 Report

The authors addressed successfully my comments and I do not see other new doubts regarding the manuscript content. In my opiion, the paper is now ready to be published.